# Synthesis, Characterization and Cytotoxicity Studies of Aminated Microcrystalline Cellulose Derivatives against Melanoma and Breast Cancer Cell Lines

**DOI:** 10.3390/polym12112634

**Published:** 2020-11-10

**Authors:** Farzana Nazir, Mudassir Iqbal

**Affiliations:** Department of Chemistry, School of Natural Sciences, National University of Sciences and Technology (NUST), Islamabad 44000, Pakistan; farzana.nazir@sns.nust.edu.pk

**Keywords:** cellulose, MCC, tosylated cellulose, aminated cellulose, cytotoxicity, melanoma, breast cancer cell lines

## Abstract

Cellulose based materials are emerging in the commercial fields and high-end applications, especially in biomedicines. Aminated cellulose derivatives have been extensively used for various applications but limited data are available regarding its cytotoxicity studies for biomedical application. The aim of this study is to synthesize different 6-deoxy-amino-cellulose derivatives from Microcrystalline cellulose (MCC) via tosylation and explore their cytotoxic potential against normal fibroblasts, melanoma and breast cancer. 6-deoxy-6-hydrazide Cellulose (Cell Hyd) 6-deoxy-6-diethylamide Cellulose (Cell DEA) and 6-deoxy-6-diethyltriamine Cellulose (Cell DETA) were prepared and characterized by various technologies like Fourier transform infrared spectroscopy-attenuated total reflectance (FTIR-ATR), nuclear magnetic resonance spectroscopy (NMR), X-ray diffractogram (XRD), Scanning Electron microscopy (SEM), Elemental Analysis and Zeta potential measurements. Cytotoxicity was evaluated against normal fibroblasts (NIH3T3), mouse skin melanoma (B16F10), human epithelial adenocarcinoma (MDA-MB-231) and human breast adenocarcinoma (MCF-7) cell lines. IC_50_ values obtained from cytotoxicity assay and live/dead assay images analysis showed MCC was non cytotoxic while Cell Hyd, Cell DEA and Cell DETA exhibited noncytotoxic activity up to 200 μg/mL to normal fibroblast cells NIH3T3, suggesting its safe use in medical fields. The mouse skin melanoma (B16F10) are the most sensitive cells to the cytotoxic effects of Cell Hyd, Cell DEA and Cell DETA, followed by human breast adenocarcinoma (MCF-7). Based on our study, it is suggested that aminated cellulose derivatives could be promising candidates for tissue engineering applications and in cancer inhibiting studies in future.

## 1. Introduction

Cancer is the uncontrolled cell division of abnormal cells clinically known as the malignant tumor or malignant neoplasm. Cancer is the second deadliest disease after heart disease, resulting in five hundred and fifty thousand deaths per year, 7.6 million in 2008 and expected to be 13.1 million by 2030 [1]. At present, drug resistance, side effects like systemic toxicity, high mortality rate and high costs [2] limit many cutting edge researches and chemotherapeutics. The medicinal chemistry field is always looking for the development of new anticancer therapeutics with lower toxicity and high efficiency [1].

Polymeric materials from renewable resources are widely used in medical, food, agriculture, biomedicinal and environmental studies. Natural polymers such as polysaccharides play the most efficient role in biomedical product preparation. Polysaccharides have wide molecular weight range and good number of functional groups for suitable chemical modification. Polysaccharides positively affect the pharmacokinetics and pharmacodynamics of enzyme molecules, proteins, and small drugs. Most widely studied polysaccharides are cellulose, starch, chitosan and chitin [3].

Cellulose is the most abundant polysaccharide [4] widely used in environmental, industrial and medical uses [5]. Cellulose has attractive properties such as good mechanical properties, low price, renewability [6], wide availability, recyclability and biodegradability, making it an interesting candidate for innovative applications in technical and biomedical areas. Inter and intra molecular hydrogen bonding between glucose units makes cellulose insoluble in common solvents. To resolve this problem, chemical or physical modification of cellulose becomes a necessity [7]. Chemical functionalization [8] or grafting polymers [9] on cellulose is one of the most adopted ways to use cellulose extensively. Prescence of abundant active hydroxyl groups makes possible functionalization of cellulose by different functional groups like nitramines, azides, tetrazoles, triflate and silanes. To achieve the chemical modification of the cellulose, it is important to activate the hydroxyl group to increase its leaving ability. Among various chemical activating agents, p-toluenesulfonyl chloride is a prominent reagent to employ nucleophilic substitution reactions (SN) for the modification of cellulose. Tosylation has been widely adopted method for functionalization of cellulose to attach small molecules because these reactions are easy to handle and cost effective [10,11]. Tosylated cellulose is helpful for cellulose utilization, by converting it into 6-deoxy-amino-cellulose [12] and 6-deoxy-6-azidocellulose [13], very important precursors in click chemistry.

The amino group containing cellulose has wide applications in household, water treatment and biomedicine [5]. Microcrystalline cellulose (MCC) was amine functionalized by using (3-chloropropyl) triethoxysilane for heavy metal removal from water [14]. Cellulose derivatives containing reactive amino groups immobilize enzyme/disfunction enzyme helpful in biosensor technology [15]. Amino Cellulose has been studied for biomimetic catalysis [16]. Amino cellulose has been used commercially, having antibacterial properties [17]. Hypergrafed cellulose was prepared by bis(2-chloroethyl)amine and soluble cellulose tosylates reaction for potential natural antimicrobial agent against various gram negative and positive bacteria [17]. Amino tosyl cellulose prepared with tosyl celluloses with ethylenediamine and 4-chlorobenzylamine was subjected to nanoprecipitation to obtain nanoparticles. Nanoparticles were FITC labelled to explore the potential of biomolecules for the development of fluorescence immunoassay [18]. Amino cellulose can be transformed into nanoparticles of 50 to 180 nm, which may be applied as biocompatible sensors in living cells and as carriers for drugs in medical applications.

To use MCC for biomedical applications, the cytotoxic profile of MCC and its derivatives must be taken into consideration. MCC showed anti-inflammatory response against microphages THP-1 by significantly attenuating the TNF-α expression [19]. MCC from wild grass *Setaria glauca* was incorporated with isoniazid drug for drug delivery. MCC showed no cytotoxicity in hemolytic assay [20]. Carboxymethyl cellulose with amino phenylpropanoic acid was synthesized, and its cytotoxic index (IC_50_) for MCF-7 breast cancers was evaluated [21]. Chemical modifications like amination may lead to change in interaction of cellulose towards the different cell lines resulting in desired biomedical application [22].There is an increasing interest in exploring the cytotoxic effect of the modified amino nanocellulose (NC). For example, Guo et al. functionalized nanocellulose with pegylated metal chelating polymers for drug delivery purpose and studied against human ovarian cancer cell line (HEYA8) [23], yet in another study, Cationic Poly(2-aminoethylmethacrylate) and Poly(*N*-(2-aminoethylmethacrylamide) Modified Cellulose Nanocrystals were tested against MCF-7 (human breast adenocarcinoma cells) [24]. Tris(2-aminoethyl) amine functionalized nanocrystalline cellulose prepared by C_TOS_ intermediate was coated on magnetic nanoparticles for drug delivery of methotrexate. These nanoparticles were found highly cytotoxic in MCF7 breast cancer cell line after 48 h incubation [25]. In case of microcrystalline cellulose (MCC) amine derivatives limited data is available regarding the cytotoxic effects of cells. El-Sayed et al. carried out cytotoxic studies against BJ1 of 3-aminopropyltrimethoxysilane and TiO_2_ grafted tosylated cellulose [26]. Therefore, herein, we functionalized (MCC) with different amine derivatives via tosylation strategy. Further, cytotoxic effect of amine functionalized cellulose derivatives has been evaluated in both normal and cancer cell lines.

## 2. Materials and Methods

### 2.1. Materials

All chemicals and reagents, viz., Cellulose, microcrystalline (MCC) (DAEJUNG 20–100 μm), p-Toluenesulfonyl chloride (DAEJUNG), Lithium Chloride Anhydrous LiCl, (DAEJUNG), *N*,*N*-Dimethyl acetamide synthetic grade (DMA) (Merck), Triethylamine Hydrazinium hydroxide (TEA) (Merck), Diethylamine (DAEJUNG), Diethylenetriamine (SAMCHUN), Ethanol (Merck) and Deionized water (Sigma Aldrich) were procured. MCC was dried at 100 °C for 5 h before being used for synthesis reactions. All other reagents were of synthetic grade and were used without further purification. Dulbecco’s Modified Eagle Medium (DMEM), TrypLE Express Enzyme (1×), Penicillin-Streptomycin (10,000 U/mL) Fetal Bovine Serum (FBS) and phosphate buffered saline (PBS) were obtained from (Gibco, Thermo Fisher Scientific Inc., Waltham, MA, USA). NIH3T3, B16F10, MDA-MB-231 and MCF-7 cell lines were obtained from American Type Culture Collection (ATCC), (Manassas, VA, USA).

### 2.2. Method

#### 2.2.1. Tosylation of Cellulose

Dried microcrystalline Cellulose MCC, 10 g (61.6 mM of anhydrous glucose unit (AGU)) was taken in 1000 mL flask, and 250 mL of *N*,*N*-Dimethyl acetamide synthetic grade (DMA) was added and stirred at 120 °C for 1 h. Resulting slurry was cooled to 100 °C, and then, Lithium Chloride Anhydrous LiCl, (20 g) in DMA (50 mL), was added dropwise. Stirring was continued overnight until the complete dissolution of the Cellulose. A volume of 37.2 mL of Triethylamine (TEA) (370 mM), 6 mol/AGU in 20 mL of DMA, was added to the cellulose solution at constant stirring for half hour. The solution temperature was dropped from room temperature to 3–8 °C, and p-Toluene sulfonyl chloride (70.6 g, 369.6 mM, 6 mol/AGU dissolved in 50 mL of DMA) was added slowly. Reaction was continued on stirring for 24 h, and color of the mixture changed from yellow to dark orange. Homogenous mixture was poured into ice water. Precipitates were collected by filtration and then washed by deionized water (8 L) and 1 L of ethanol, suspended in 500 mL of boiling acetone and recollected in deionized water. After filtration, it was washed with Ethanol (500 mL × 3). Tosylated Cellulose obtained was dried below 50 °C in oven.

Tosylated Cellulose (C_TOS_): White powder = 7.03 g, yield 77%, Elemental Analysis (%age Found); C = 46.05, H = 4.94, S = 5.01, DS (Degree of Tosylation) = 0.48 (based on sulfur analysis), FTIR cm^−1^: 3421 (ⱱ_str_OH) 2887 (ⱱCH), 1619 (ⱱ_asym_C–O–C), 1594 (ⱱ_arom_C–H or), 1358 (ⱱ_assym_SO2), 1172 (ⱱ_sym_SO2), 812 (ⱱ_arom_C–H) 1358, ^1^HNMR (400MHz DMSO-d6) δ ppm: 7.75 (d, 2H, ortho), 7.35 (d, 2H, para), 4.94–3.99 (m, 1H, CH), 3.90–3.18 (m, 5H, CH), 3.14–3.00 (m, 1H, CH), 2.42 (s, 3H, tosyl CH_3_). ^13^C NMR (100 MHz, DMSO) 138.07 (aromatic), 128.51 (aromatic, meta), 125.95 (aromatic, ortho), 110.31 (anomeric), 84.84, 81.07, 75.32, 73.10, 65.87, 21.59 (tosyl CH_3_).

For Degree of substitution (DS) of tosylated cellulose was calculated by using Elemental analysis for sulfur. Formula for the calculation of the degree of substitution DS = M Glucose units × S(%)/M Sulfur × 100(%) − M Tosylated Cellulose × S(%) [27].

#### 2.2.2. Subsitution of C_TOS_ with Amines to Synthesize 6-Deoxy-Aminocellulose

A series of cellulose amine derivatives were produced by substitution of tosyl group. Tosylated cellulose (C_TOS_) (2 g) was dissolved in 2 ml of DMF with constant stirring. A total of 25 equivalents [28] of either Hydrazinium hydroxide, Diethylamine or Diethylenetriamine per AGU in 10 mL of DMF having 1 ml of TEA were drop wise added to viscous solution of C_TOS_ with constant stirring at 90 °C. Reaction was continuously refluxed at 90 °C for 24 h while progress was monitored by TLC. Reaction mixture was dialyzed in deionized water and ethanol to obtain the product. Obtained derived polymeric product was collected by filtration and dried at 50 °C under vacuum.

6-deoxy-6-hydrazide Cellulose (Cell-Hyd) Tea pink powder = 1.69 g, yield 84%, Elemental Analysis (%age Found), C = 41.34, H = 6.64, N = 7.95. FTIR cm^−1^: 3294, 3330 (broad ⱱ_str_N–H,–OH) 2889 (ⱱCH), 1650 (ⱱ_arom_C–H), 1370 (ⱱCH/CO), 2201 (ⱱCN), 1608 ( C=N aromatic), 1589 (ⱱ_bend_N–H) 1170 (ⱱ_str_C–N) cm^−1^, ^1^HNMR (400MHz DMSO-d6) δ ppm; 6.20 (s, –NH), 5.42 (d, anomeric H of Cellulose), 5.15–5.09 (d, OH of Cellulose), 5.03–4.96 (d, OH of Cellulose), 4.69 (s, OH of Cellulose), 4.36–4.20 (m, H of Cellulose), 3.91–3.46 (m, H of Cellulose), 3.23 (s, –NH_2_), 3.10–3.00 (m, H of Cellulose). ^13^C NMR (100 MHz, DMSO) 109.19 (anomeric), 82.31, 79.57, 76.19, 69.67.

6-deoxy-6-diethylamide Cellulose (Cell-DEA) Off white powder =1.6 g, yield 80%, Elemental Analysis (%age Found), C = 46.29, H = 6.71, N = 3.25.FTIR cm^−1^: 3362 (broad ⱱ_str_N–H,O–H) 2883 (ⱱC–H), 1659 (ⱱ_arom_C–H) cm^−1^, 1375 (ⱱ_asymml bend/str_C–H), 1166 (ⱱ_str_C–N) cm^−1^, ^1^HNMR (400 MHz DMSO-d6) δ ppm; 5.23 (d, anomeric H of Cellulose), 5.03–4.94 (d, H of Cellulose), 4.68 (s, H of Cellulose ), 4.37–4.22 (m, H of Cellulose), 3.92–3.43 (m, H of Cellulose), 3.12–3.01 (m, H of Cellulose), 2.59 (m–CH_2_–), 1.15 (t, 6H, –CH_3_). ^13^C NMR (100 MHz, DMSO) 101.74 (anomeric), 81.37, 79.16, 67.07, 62.63, 52.44 (–N–CH_2_–), 7.95 (–CH_3_).

6-deoxy-6-diethyltriamide Cellulose (Cell-DETA) Brown powder =1.78 g, yield 89%, Elemental Analysis (%age Found), C = 44.73, H = 7.24, N = 9.55. FTIR cm^−1^: 3337 (broad ⱱ_str_N–H, O–H), 3288, 2897–2878 (ⱱCH), 1650 (ⱱ_arom_C–H), 1570 (ⱱ_ben_ NH), 1471 (ⱱ_asymm,ben_C–H), 1152 (ⱱ_str_C–N), 1312 (ⱱ_asymm,ben_C–H), 1035 (ⱱ_str_C–O) cm^−1^, ^1^HNMR (400 MHz DMSO-d6) δ ppm; 4.39–4.20 (m, H of Cellulose), 3.74–3.40 (m, H of Cellulose), 3.34–3.19 (m, H of Cellulose), 3.18–3.06 (m, H of Cellulose), 3.09–2.98 (m, H of Cellulose), 3.7 (s, –NH) 2.59–2.69 (m, 8H, CH_2_), 1.77 (s, –NH–), 1.03 (s, –NH_2_). ^13^C NMR (100 MHz, DMSO) 104.76 (anomeric), 102.84 (anomeric), 81.99, 75.13, 70.15, 69.12, 65.73, 57.18 (–CH_2_–CH_2_–NH–CH_2_–CH_2_–), 48.61, 46.79, 37.90 (–CH_2_–CH_2_–NH–CH_2_–CH_2_–).

### 2.3. Measurement

FTIR spectra were recorded on Bruker attenuated total reflectance Fourier transform infrared (ATR FT-IR) spectrophotometer (Bruker platinum ATR model Alpha spectrophotometer, Karlsruhe, Germany). Twenty mg of samples was scanned over 4000 to 400 cm^−1^ wave number. ^1^HNMR spectra were recorded for all samples at room temperature in deuterated dimethyl sulfoxide (DMSO-d6) on a 400 MHz Bruker AV400 spectrometer (Bruker Corporation, Billerica, MA, USA) with 64 scans for concentration of 20 mg/mL^−1^. SEM Morphological studies of the powdered derivatives were checked on SEM (JSM-64900) with (EDX) spectrometer (JEOL, Akishima, Japan). Samples were sputter coated with iridium using South Bay Technology Ion Beam Sputtering/Etching System. After sputter coating, sample morphology was observed under SEM Elemental analyses were carried out on a CKIC 5E-CHNS-2200 and CKIC5E-IRS II ultimate analyzer (CHANGSHA KAIYUAN INSTRUMENTS CO LTD., Changsha, China). X-ray Diffraction patterns of cellulose and its derivatives were recorded with a X-ray diffractometer (D8 advance BRUKER, Billerica, MA, USA) in transmission mode over 2θ ranges 5−80° with Cu Kα radiation. Phase conformation of the synthesized nanomaterials was also confirmed at 40 kV voltage and 40 mA current while the instrument uses Cu Ka radiation (λ = 1.54060 Å) for X-ray production. Zeta potential ζ Zeta potential was measured on a Zetasizer NanoS Series (Malvern Panalytical Ltd., Worcestershire, UK). Samples 1 mg/10 mL in deionized water were suspended using a sonication bath for half an hour before testing in a disposable folded capillary cell at 37 °C. Furthermore, the samples were incubated at 37 °C for 48 h under constant stirring before recording the zeta potential measurements to check the stability of zeta potential. The ζ- potential plot was analyzed using the Zetasizer software PCS V.1.4 (Malvern Instruments, Malvern, UK). Swelling ratios of the samples were determined by taking 10 mg sample in pre weighed Eppendorf tubes with 1 ml of Dulbecco’s phosphate-buffered saline (DPBS). After 24 h, DPBS was removed from samples, and samples were again weighed. Swelling ratios were calculated by following formula (weight of the wet sample/weight of dry sample × 100).

Cell culture In vitro cytotoxicity of the synthesized cellulose derivatives was evaluated against 4 cell lines. One fibroblast cell line NIH3T3 (mouse embryo fibroblasts) and three cancer cell lines B16F10 (mouse skin melanoma), MDA-MB-231(human epithelial adenocarcinoma) and MCF-7 (human breast adenocarcinoma) were used to calculate the IC_50_ values. Frozen Cell lines were passaged in DMEM supplemented with 10% FBS and 1% Penstrep five times at 37 °C and humidified air with 5% CO_2_. Cell viability and cytotoxicity index IC_50_ values were determined by seeding the 96 well plates at concentration 30,000 cells/mL or 10,000 cells/cm^2^ for each cell line and incubating at 37 °C in humidified air with 5% CO_2_. After 24 h, media was removed, and cells were washed with PBS to remove any dead cells. Then, 100 ul of serial diluted concentrations of cellulose and its derivatives (5, 10, 20, 40, 60, 80, 100, 150, 200, 250, 500 and 1000 μg/mL) was added to each well and incubated in the same condition for the next 24 h. Cellulose and its derivatives were dissolved in DMSO (maximum concentration of DMSO in the 0.5% per well) and then diluted in DMEM. Untreated cell wells were only media added to serve as control and were incubated for 24 h. Media was removed after 24 h, and cells were incubated for 2 h with PrestoBlue Cell viability Reagent (10% PrestoBlue in DMEM). Cell viability and cytotoxicity index IC_50_ was calculated by reading the plate on a plate reader at fluorescence intensity excitation/emission 535–560/590–615. Triplicate samples were used for analysis. Prism-GraphPad-5 was used to draw the IC_50_ Curves with standard errors, and then, IC_50_ values were calculated. All data were presented here as an average ± standard deviation (S.D.), and *p* < 0.05 considered as significant difference. All cell experiments were performed with triplicate samples in 3 independent experiments. Prism-GraphPad 5 was used to analyze the results and create figures. Nonlinear regression curve fit (Dose–response) was used to plot the cytotoxicity results of each compounds thereby. IC_50_ values were obtained. The obtained IC_50_ values were subjected to one-way ANOVA analysis using an appropriate Tukey to compare all pairs of columns test for statistical comparison. In figures, statistical significance was defined as * *p* ≤ 0.05, ** *p* ≤ 0.01, *** *p* ≤ 0.001. Live/Dead Assay was carried out at 24h stage according to the protocol for LIVE/DEAD^®^ Viability/Cytotoxicity Kit (mammalian cells). Cells were cultured according to same protocol as above at IC_50_ value for 24 h. Untreated cells were used as control. Cells were washed with PBS and then Calcein AM (1 µL/mL of 50 µM stock) Ethidium homodimer-1 (2 µL/mL of the 2 mM stock) 50 ul was added to each well. Plate was incubated for 30 to 40 min, and then, the cells were observed on inverted fluorescence microscope (Zeiss Axio Observer Z1, Zeiss, Germany) at 494/517 nm for Calcein-AM (live cells as green indicating intracellular esterate activity) and at 528/617 nm ethidium homodimer (dead cells as red indicate loss of plasma membrane activity). Number of live cells were counted by Image J (US National Institutes of Health, Bethesda, MD, USA) software and from four random fields of 3 replicates of samples. Live cell percentage was counted by dividing number of live cells by total number of cells. Four random field images were taken at 10×.

## 3. Results

### 3.1. Tosylation

Tosylated Cellulose (C_TOS_) with degree of substitution (DS) 0.48 was obtained by previously reported method Rahn et al. [29]. Shortly, Cellulose was reacted with 6 equivalent of tosyl chloride by dissolving Cellulose in *N*,*N*-dimethylacetamide/LiCl with 6 equivalent Triethylamine for 24 h at 10 ℃. Lower temperature and Triethylamine [30] minimize the formation of 6-deoxy-6-chloro groups [31]. It has been reported that tosylation reaction occurs preferably at position 6 rather than positions 2 and 3 [29]. A total of 6 equivalents of p-tosyl chloride against hydroxyl group increases the DS from 0.3 to 0.5 with percentage yield corresponding to 77%. Lab synthesized Tosylated cellulose (C_TOS_) was evaluated by FTIR, NMR and Elemental analysis (for degree of substitution). Infrared spectroscopy cannot fully explicate the structure but gives useful information about the functional group. Infrared spectra of Tosylated cellulose in correlation to microcrystalline Cellulose showed shifting of band maxima and appearance of new band positions. Pristine microcrystalline cellulose peaks are 3333 (ⱱ_str_OH), 2899 (ⱱ_sym_CH) and 1633 (ⱱ_asym_C–O–C). Other peaks at 1317 are for ⱱ_bend_CH_2_ and ⱱ_wagg_ vibrations. [32] C–H/C–O bending vibration of polysaccharide ring appeared at 1366 cm^−1^, C–O–C ß-glycosidic linkage at 1158 cm^−1^, C–O stretching at 1022 cm^−1^ and COH stretching vibration at 897 cm^−1^. C_TOS_ absorption bands 3422 (ⱱ_str_OH) 2887 (ⱱCH), 1619, 1594 (ⱱ_arom_C–H), along with tosyl SO_2_ group appeared at 1358 (ⱱ_assym_SO_2_), 1172 (ⱱ_sym_SO_2_), 812 (ⱱ_arom_C–H) cm^−1^ conforming the tosylation of Microcrystalline Cellulose (C_TOS_) [33]. Solubility of the prepared tosylated Cellulose (C_TOS_) was observed. Obtained tosylated Cellulose (C_TOS_) was soluble in aprotic highly polar organic solvents (DMSO) and was insoluble in a lot of protic or polar solvents (like water, Ethanol). Elemental analysis was helpful for finding the degree of substitution by finding out the percentage of the elements found in the sample. Using Sulfur content in the samples Degree of substitution (number of hydroxyl groups substituted per Glucose unit) was carried out using the formula [10]. ^1^HNMR (solution of C_TOS_ in DMSO-d_6_) shows the tosyl group hydrogens appear as two broad and two sharp peaks for the benzene ring between 7 and 8 ppm 7.75 CH, 7.35 CH and methyl group CH_3_ appears at 2.42 ppm [5]. Peaks for the Cellulose, which are between 3.5 to 5.5 ppm, are for the cellulose polymer backbone. The intensity of the ^1^HNMR for tosylate group covalent bond to Microcrystalline cellulose backbone was not very high in earlier attempts, suggesting that DS_TOS_ is less, meaning not all the hydroxyl groups can be substituted; rather, degree of substitution depends on the ratio of Tosyl Chloride to Cellulose in reaction mixture. Keeping this in mind, we modified the method previously reported to obtain the tosylated cellulose up to 0.5 DS. Degree of substitution was also responsible on the solubility of the Tosyl Chloride and the products formed [34]. Solubility is prerequisite for further reaction and characterization such as ^1^HNMR. Tosylated cellulose with DS_TOS_ more than 0.43 are DMSO soluble; upon DS_TOS_ 0.95, it is further soluble in aprotic solvent in DMSO, DMA and DMF. Tosylated cellulose was prepared following Rahn et al.’s synthetic method of dissolving Microcrystalline Cellulose in DMA/LiCl; having DS_TOS_, 0.46 are soluble in DMF, DMSO and DMA [29]. Our lab-synthesized Tosylated Cellulose was DMF soluble. No hints of any side reaction were found.

### 3.2. Synthesis of Microcrystalline Cellulose Amine Derivatives

Tosylated cellulose (C_TOS_) was converted to amine cellulose derivatives by stirring tosylated cellulose with corresponding amines in a 1:25 per anhydroglucose unit (AGU) mole ratio. Excess amount of amine (1:25) ratio was used to avoid any side reaction and crosslinking and obtain a soluble product [18,28]. Reaction was preceded for 5 h at 100 °C in DMF as reported for tosylated group replacement by many amines to form 6-deoxy-6-substituted amine cellulose derivatives. Amine cellulose derivatives of C_TOS_ are regioselective modification at position 6. Nucleophilic substitution is preferred at primary tosylated sites rather than secondary tosylated [35]. Nucleophilic substitution reaction is not quantitative in all reactions [36], and because of polymer chemistry, it is impossible to remove the tosyl group as in case of other organic compounds of lower molecular weight. Reaction products of higher DS_TOS_ microcrystalline cellulose also show unremoved tosyl groups from products as tosyl groups are present at position 2 of Microcrystalline cellulose [37].

In the present study, tosylated cellulose was reacted with Hydrazide, diethylamide and diethyl triamine. Yield of the aminated cellulose derivatives Cell Hyd, Cell DEA and Cell DETA was found to be 84%, 80% and 89%, respectively. Figure 1a shows the FTIR spectra as compared to the Microcrystalline Cellulose. From these spectra, it is indicated that the amines have been substituted on the Microcrystalline Cellulose backbone. FTIR spectra of aminated Cellulose derivatives were in accordance with already reported data as described by [38].

FTIR spectrum for (Cell Hyd) the stretching peaks for the N–H can be seen at 3300 and 3294 cm^−1^ because of the decoration of Hydrazide on Microcrystalline Cellulose backbone with an over tone appearing N–H (bend primary amines) at 1589 cm^−1^ along with the characteristic peaks of the Microcrystalline Cellulose such as 2883 (ⱱCH), 1659 (ⱱ_arom_C–H) cm^−1^, 1375 (ⱱ_asymml bend/str_CH). One hydrogen of hydrazine is being used for the bonding with the Carbon evidenced from the C–N peak can be observed at 1170 cm^−1^. For Cell DEA, the N–H peaks at 3288 cm^−1^ is not existing while the O–H peak at 3362 cm^−1^ intensity is low as well as it is less broadly relevant to Microcrystalline Cellulose. Addition of the DEA to the cellulose back bone is also causing appearance of the C–N stretch peak at 1166 cm^−1^ while the CH asymmetrical is bending/stretching at 1375 cm^−1^. In Cell DETA FTIR spectrum, the characteristic 2 bands for amine of N–H 3342 cm^−1^ and 3288 cm^−1^, C–H symmetric and asymmetric at 2897 cm^−1^ and 2878 cm^−1^, N–H bending or overtone appear at 1654 cm^−1^ and 1570 cm^−1^, CH_2_ symmetric and asymmetric bending at1471 cm^−1^ 1312 cm^−1^, C–N for aliphatic amine appeared at 1160 cm^−1^ along with the signature peaks of Microcrystalline Cellulose. As mentioned in the literature [39], primary amines show overtone between 1590–1650 cm^−1^ while secondary amines show overtone at 1550 to 1650 cm^−1^. This strengthens the idea that one of the NH_2_ of hydrazine and triethylamine is bonded to the cellulose while dimethylamines N–H participated in bonding as shown in the Figure 2. It is quite attractive to mention that C–H stretching band has shifted to a higher wavenumber value from 2899 cm^−1^ along with the increase in intensity of CH_2_ bending band from 1428 cm^−1^ resulting in Cell derivatives that are more crystalline as compared to the C_TOS_ [20]. N–H wag appears (primary and secondary amines only) from 910–665 cm^−1^. 

^1^HMNR spectra of the 6-deoxy-6-substituted amine/aminated cellulose in DMSO-d6 are represented in Figure 1c Microcrystalline cellulose backbone aromatic ring hydrogens peaks appeared between 3 to 5.3 ppm [40]. For Cell Hyd, along with the MCC backbone peaks, single H of NH appeared sharply at 6.2, while NH_2_ Hydrogen expected to appear at 3.23 ppm. In case of Cell DEA [41] along with MCC peaks, CH_2_ multiplet and CH_3_ triplet appeared at 2.59 and 1.15 ppm, respectively, evidencing the successful substitution on cellulose. For Cell DETA along with MCC back bone, NH_2_ singlet appeared at 1.03, multiplet for CH_2_ Hydrogen appeared at 2.59–2.69 ppm, NH hydrogen signal appeared at 1.77 ppm, and at 3.7 ppm, there was broad signal for NH, respectively.

From the ^1^HMNR data, it is clear there is complete removal of the ionic tosyl moieties from all the Cellulose amine derivatives. From the spectra, it is clear there was successful reaction between the tosylated cellulose and amines, resulted by removal of the one hydrogen from the primary amine NH_2_ or secondary amine NH and bonding with the position 6 methylene group of MCC as is evidenced by FTIR. In case of Cell Hyd, one Hydrogen of hydrazine NH_2_ was replaced by bonding with methylene of MCC; thus, single Hydrogen of NH appeared sharply at 6.2, and second hydrazine NH_2_ Hydrogen expected to appear at 3.2 ppm was masked under the area of Cellulose backbone hydrogens. For Cell DEA, there was no peak for NH observed, suggesting the bonding occurred between Nitrogen and methylene of MCC. Cell DETA synthesis via primary amine NH_2_ and methylene carbon was vivid from the broad peak at 3.7 ppm (masked by Cellulose backbone peaks area), multiples of CH_2_ hydrogen at 2.59–2.69 and appearance of NH Hydrogen at 1.77. Appearance of some of the tosylated moieties attached to the cellulose backbone as two broad and two sharp peaks for the benzene ring between 7 and 8 was very low as reported by Heinze T et al. [5]. This can be attributed to the steric hinderance of the amine groups of DEA and the polymeric structure of Cellulose resulting in less Degree of Amination. ^13^CNMR no detectable peaks of tosyl group between 128 to 140 ppm were found while there were high intensity peaks for CH_2_NH_2_ appeared around 52 and 57 ppm while the methyl and methylene Carbons next to Hydrogen appeared between 45 to 35 ppm.

Elemental analysis was carried out for the DS_TOS_ and DS_Amine_. Degree of amination was found to be 0.40 to 0.43, suggesting that the maximum number of tosyl group has been removed. Elemental analysis data in Table 1 proved that there is no chlorine moiety resulting from a side product. No polymer degradation resulted in the synthesis reaction. Yield of the reactions was more than 80%. In all the derivatives, a trace amount of sulfur of 0.5%, 1.01% and 0.77% for Cell Hyd, Cell DEA and Cell DETA was found, respectively. Complete removal of Tosyl group was not possible because of the polymeric structure of the Cellulose, but Nitrogen content of 8.26%, 3.68% and 9.55% in case of Cell Hyd, Cell DEA and Cell DETA was found, respectively, of the derivatives, which was quite high, reflecting the substitution of the Tosyl group and modification of cellulose backbone by amines. 

To analyze the effect of the amine modifications on cellulose backbone, the crystal structure of the cellulose and its aminated Cellulose derivatives was observed with wide angle powder X-ray diffractogram shown in Figure 1b. From the data, it was possible to evaluate qualitatively that microcrystalline cellulose has the diffraction pattern of semi crystalline cellulose type I, [42] Characteristic peaks were observed at 2θ = 15.43°, 16.21°, 22.4° and 34.7° corresponding to the monolithic cellulose type I crystallographic planes (110), (110), (200) and (004) as in other studies [30,43]. The diffraction pattern of the cellulose changed as reported in previous studies [26] after the tosylation and derivatization indicating that all the chemical modification dissolution of Cellulose [44] in LiCl/DMA/TEA affected the lattice and changed the crystallinity of the cellulose as appeared from reasonable lowering of the peak intensity. C_TOS_ displayed a broad peak at 2θ = 20° (021) of amorphous cellulose II [33,45]. All aminated Cellulose derivatives (Cell Hyd, Cell DEA and Cell DETA) show decrease in the crystallinity as evident from the low peak heights [46] and more of amorphous structure as reported by others [17]. Decrease in crystalline structure from starting Microcrystalline Cellulose during functionalization is because of the disturbance of cellulose chains inter and intramolecular hydrogen bonding [47,48,49]. This indicated the correlation between the findings of the FTIR, ^1^HNMR and elemental analysis. Several other researchers [50,51] also mentioned the intense hydrogen bonding responsible for the crystalline part of cellulose, so loss of OH group is responsible for decrease in crystalline structure. From the Cell Hyd, Cell DEA and Cell DETA diffraction spectra, peaks at diffraction angles 2θ = 20.6°, 22°, 34.8°, 2θ = 20.5°, 22.1°, 36.5° and 2θ = 20.6°, 22.3°, 35.1°, respectively, show the coexistence of the Cellulose I and Cellulose II [52]. It is interesting to mention that the amine cellulose derivatives have higher peak height meaning crystallinity than the C_TOS_ indicating that the nucleophilic substitution of the Tosyl group by amines have resulted into products with increased the number of unreacted OH and NH moieties available for inter and intramolecular hydrogen bonding leading to in increased ordered compact arrangement. It is quite intriguing to mention that the XRD pattern are quite convincing with already reported work [53]. Increasing randomness of the amorphous phase in the cellulose amine derivatives caused long range spacing of polymeric chains resulting in the decrease in crystalline structure as reported elsewhere [54].

Zeta potential ζ of the cellulose and its amine cellulose derivatives was measured as shown in listed in Table 2 at neutral Ph. From Figure 1d, we observed all the derivatives shifted to a positive value of zeta potential from a negative zeta potential of cellulose as reported elsewhere [55]. Aminated cellulose derivatives expressed cationic character in aqueous solution. Zeta potential values showed a very minor change after incubation for several hours at 37 °C showing stable zeta potential values. Zeta potential values helped in the confirmation of the modification of the microcrystalline cellulose. Positive zeta potential values are because of the protonation of the amine groups. On comparing the zeta potential values of the cellulose, amine derivatives increased number of primary amino groups resulted in more positive zeta potential values [56].

Positively charged amine derivatives foster muco-adhesion resulting in increased residence time on mucosa, which is absent in the native Microcrystalline cellulose. This kind of bio-adhesion is very important in case of biomedical application. Cellulose derivatives exhibiting positive charge due to secondary amine structure have been reported in previous studies [57]. 

SEM micrographs helped in understanding the effects of all the modification on surface morphology. Microcrystalline Cellulose (MCC) exhibits the homogeneous structure of the fibers having a well oriented network like structure (Figure 3), while C_TOS_ showed a sheet-like heterogenous structure with rough surface interconnected to form porous structures. This indicates the loss of cellulose surface order after chemical modification [58]. For aminated cellulose derivatives, we see that the interconnected structure of fibers and some rough surface and sheet like porous structure appear indicating the presence of the tosyl group and amine moieties on the surface of the aminated cellulose derivatives as indicated by the ^1^HNMR [27].

SEM structure of the chemically modified microcrystalline cellulose fiber has become rough surfaced small fibers and particles rather than the clear smooth ones because during the modification, Microcrystalline cellulose has undergone swelling, new crystal structural modification with addition of the functional groups as predicted by the XRD structure [59]. Similarity type of the irregular rod like rough structure between the microcrystalline cellulose and aminated cellulose, indicating that there was homogenous amination of the polymer without much alteration of surface morphology [48]. Another useful information was acquired by EDX data as shown in Figure 3. The EDX spectra reinforced the data from Elemental Analysis by showing the presence of Nitrogen in the aminated Cellulose Derivatives in enough quantity to prove the successful synthesis of the aminated Cellulose derivatives from Microcrystalline Cellulose (MCC) via tosylation reaction. 

Swelling ratios of the Cellulose and its aminated cellulose derivatives are important because these help in understanding the solute diffusion as well as surface properties of the compounds. Figure 4f shows the swelling ratios in PBS. Cellulose showed the highest swelling ratio at 69 ± 2, which decreased in all other aminated cellulose derivatives. This relatively high cellulose swelling ratio is because of the penetration of the water in amorphous regions of cellulose causing changes in the crystal structure of the cellulose from Cellulose I to Cellulose II as corroborated by XRD. Aminated Cellulose derivatives, because of presence of the NH_2_ moieties with the cellulose backbone and disordered arrangements of the polymeric chains, caused the high swelling ratios 60.8 ± 1, 50 ± 2 and 59.8 ± 3 for Cell Hyd, Cell DEA and Cell DETA, respectively.

### 3.3. Biological Evaluation

#### 3.3.1. Cytotoxicity

Cellulose amine derivatives have been evaluated against bacteria in many reports, but there has been less attention given to the cytotoxicity. To ensure the use of cellulose amine derivatives in biomedical applications, it is very important to explore the cytotoxicity of these materials as a first step. Literature survey showed Finger S et al. reported 6-deoxy-6aminoethyleneamino cellulose with different degree of substitution (DS) via tosylation against keratinocyte cell (HaCaT). Their studies concluded that lowest DS (0.3) was most biocompatible with (HaCaT) [60]. While in another study, El-Sayed et al. carried out cytotoxic studies against BJ1 of 3-aminopropyltrimethoxysilane and TiO_2_ grafted tosylated cellulose. TC DS_TOS_ = 0.77, TC-Si, TC-Si/TiO_2_ were prepared. TC TC-Si showed moderate cytotoxicity by 17% and 23.8% while TC-Si/TiO_2_ enhanced the proliferation of BJ1 by 42% [26]. In this study, novel MCC derivatives Cell Hyd, Cell DEA and Cell DETA have DS_amination_ 0.40–0.43. The abovementioned studies encouraged us to further explore and investigate cellulose derivatives interaction with cell lines. Breast cancer is the second most deadly cancer in the US. Melanoma is a cancer that occurs in cells (cutaneous, mucosal or ocular) responsible for formation of colored pigment melanin. Melanoma is the most abundant cancer affecting those between 25 to 29 years in the USA. These most common skin cancers representing cell lines were used for the cytotoxic study against aminated cellulose derivatives.

The ability of the synthesized aminated cellulose and microcrystalline cellulose to inhibit the metabolic activity of the cancerous B16F10 (mouse skin melanoma), MDA-MB-231 (human epithelial adenocarcinoma) and MCF-7 (human breast lines adenocarcinoma) and non-cancerous cell lines NIH3T3 fibroblasts was evaluated by standard Presto blue assay for cell viability. Presto blue reagent was used to measure viability of cells. Presto blue reagent (Resazurin) was reduced to resorufin by metabolically active cells indicating mitochondrial activity of the living cells. Nontreated cells were used as control, and cytotoxic activity of compounds was evaluated as the concentration of compounds inhibiting 50% of already grown cells (IC_50_) after 24h using Prestoblue assay listed in Table 3. To ensure the proper solubility of the derivatives, amines were first dissolved in DMSO and then were diluted to DMEM + 10%FBS + 1% Penicillin; since the maximum concentration of DMSO was not more that 2%, DMSO cytotoxicity was limited.

Cytotoxicity of the compounds was evaluated by adding serial diluted concentrations of amine functionalized compounds to cells seeded in 96-well plate. IC_50_ (the half maximal inhibitory concentration) value of the compounds was calculated by dose-response curves determined from the cytotoxicity assay by plotting IC_50_ curves shown in Figure 4a–d. Interestingly, all aminated cellulose derivatives exhibited good metabolic inhibiting activity against cancer cell lines (B16F10, MDA-MB-231 and MCF-7) as compared to normal NIH3T3 cells listed in Table 3. Based on IC_50_, values of pure cellulose were not cytotoxic at lower concentrations for all cell lines, and IC_50_ values are highest for NIH3T3 263.9 μg/mL while 207, 201.6 and 238.9 μg/mL are for B16F10, MDA-MB-231 and MCF-7, respectively. For cancer cell lines, IC_50_ values ranged between 75 μg/mL and 205 μg/mL for aminated cellulose derivatives as shown in Figure 4e. Cell DEA and Cell DETA displayed higher anticancer activity as compared to Cell Hyd for melanoma and breast cancer cell lines as shown in Figure 4e. Cell DETA showed promising results against B16F10 (mouse skin melanoma) (IC_50_ < 101.5 μg/mL) and MCF-7 (human breast adenocarcinoma) cell line (IC_5__0_ < 74.92 μg/mL) while Cell DEA was more cytotoxically potent against MDA-MB-231 (human epithelial adenocarcinoma) (IC_50_ < 154.6 μg/mL). All compounds showed excellent metabolic inhibiting results against B16F10 (mouse skin melanoma) Cell Hyd 130.5 μg/mL, Cell DEA103.64 μg/mL and Cell DETA 101.5 μg/mL. Same serial gradient concentrations used against cancer cell lines were screened against fibroblast NIH3T3 cell lines for selective cytotoxicity comparison. For fibroblast cells, safe concentration values were up to 231 μg/mL, 222 to 260 μg/mL for Cell Hyd, Cell DEA and Cell DETA, respectively, after that cells metabolic activity decreased. Microcrystalline Cellulose was not found cytotoxic. Abercrombie et al. reported that cancer cells may have slightly elevated negative surface charge on the cell membrane than normal cells [61]. The low metabolic activity in cancer cells is due to interaction of positive potential (zeta potential) on aminated cellulose derivatives with negative charge on cell membrane [55,62]. Thus, depolarization of membrane and imbalance in ionic transport occurred [63]. An alteration in cell signal stimulates the mitochondria to generate intracellular oxidation stress, thereby decreasing metabolic activity of cells. The significant difference in the cytotoxicity of Cell DETA could be attributed to the synergistic effect of surface potential and long chain aliphatic group on tertiary amine.

Primary amine derivatives as compared to the tertiary amine derivatives showed considerable growth inhibitory activity to the cancer cell lines as compared to fibroblast cell line. It can be seen vividly that the compounds are moderately selective towards cancer cell lines over normal cell lines. Recently, Cheng et al. developed amine functionalized anticancer drugs, and they had more selectivity between normal and cancer cells [64]. Comparing the IC_50_ values of the normal cell line and the cancer cell line, Selectivity Index can be calculated. SI is useful as indicator of the compound’s selective cytotoxic behavior for cancer cells as compared to the normal cell line. The higher the values of SI, the more selective the compound is [65].

In another report of amine functional derivatives, carrying longer aliphatic chains on the nitrogen atom is superior to shorter ones in improving cytotoxicity [64]; a similar trend can be seen in our study of the difference in the metabolic rate inhibition effect of Cell Hyd compared to Cell DETA.

#### 3.3.2. Live Dead Assay

Live/dead assay was performed to confirm the cytotoxic behavior of the aminated cellulose derivatives towards the cancer cell lines and fibroblast cell lines. Cells were treated with the IC_50_ value (derived from Table 3) and a value higher than IC_50_ (250 μg/mL) and a value lower than IC_50_ value (20 μg/mL) results to visualize the effect of the compounds on the cancer cell lines. Fluorescent visualization of live and dead cells is shown in Figure 5. Live cells are detected as green fluorescence by Calcein indicating intracellular esterate activity, and dead cells are detected as red fluorescence when ethidium homodimer binds to DNA. Ethidium homodimer is excluded from live cells because of cell membrane integrity, while in dead cells, loss of plasma membrane activity allows ethidium homodimer entering cells and binding to DNA. Live/Dead Assay revealed the live/dead cells proportion in the culture medium. It is clear from Figure 5 that the amine functionality gives aminated cellulose derivatives cytotoxic ability and causes arrest of cell proliferation. Moreover, the higher concentrations of Cell Hyd, Cell DEA and Cell DETA not only kill the maximum cancer cells within 24 h but, at the same time, their high concentration becomes cytotoxic for the normal cell lines.

Live dead assay results suggested that at IC_50_ values the morphology of the cells have changed as compared to the (20 μg/mL). Cells are making round morphologies and are detached from each other losing their intercellular network. Cells showed the shrinkage in size one of the fundamental characteristics of apoptosis [66]. Moreover MCF-7 cells showed the cluster formation. Cytotoxicity was also observed by Image J software on the live dead cell images. Percentage of viable cells found from Image J has been shown in Figure 5. Overall, the cells in the (20 μg/mL) were showing more than 91% cell viability while in the presence of aminated cellulose derivative at IC_50_, we can see the live cell percentage is decreasing significantly by 50 percent in cancer cells after 24 h, as reported elsewhere [67]. All compounds had high cytotoxicity against B16F10 cell lines. MDA-MB-231 showed almost same vulnerability to Cell DEA and Cell DETA. While Cell DETA is very effective against the MCF-7 cell lines. In Live dead assay images, NIH3T3 cells have good morphology and intercellular network suggesting non cytotoxic to normal cell lines under 200 μg/mL. This suggests that in safe concentration, Cell Hyd, Cell DEA and Cell DETA can be used for tissue engineering applications in further studies.

## 4. Conclusions

MCC aminated derivatives Cell Hyd, Cell DEA and Cell DETA were successfully synthesized by tosylation intermediate with DS_amination_ 0.40–0.43. Cell Hyd, Cell DEA and Cell DETA have C–N peak 1160–1170 and N-H peak between 3200 cm^−1^. ^1^HNMR confirmed the successful synthesis of cellulose derivatives. Dissolution and chemical modification affected the inter and intra molecular hydrogen bonding thus Cell Hyd, Cell DEA and Cell DETA have decreased crystallinity as compared to MCC in XRD and SEM. MCC was non cytotoxic for cell lines whereas positive zetapotential values of Cell Hyd, Cell DEA and Cell DETA were responsible for the selective cytotoxicity for Melanoma and Breast Cancer cell lines as compared to fibroblast cell lines (NIH3T3). The mouse skin melanoma (B16F10) is the most sensitive cells to the cytotoxic effects of Cell Hyd, Cell DEA and Cell DETA, followed by human breast adenocarcinoma (MCF-7). Cell Hyd, Cell DEA and Cell DETA are safe up to 200 μg/mL for the fibroblast cell lines (NIH3T3). Live/Dead assay provided the assessment that live/dead cells proportion at IC_50_ is almost 50%. Based on the abovementioned results, we can conclude that, Cell Hyd, Cell DEA and Cell DETA proved to be a practical candidate for further study as biocompatible scaffold in tissue engineering applications, on one hand, while on the other hand, effective anticancer agent in cancer therapeutics.

## Figures and Tables

**Figure 1 polymers-12-02634-f001:**
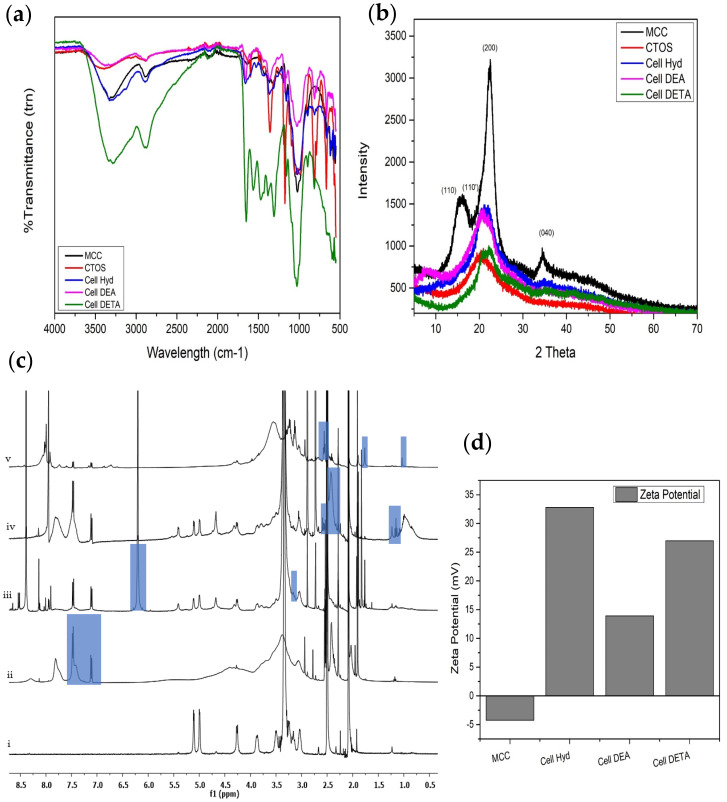
Characterization of cellulose and cellulose amine derivatives: (**a**) FTIR; (**b**) XRD; (**c**) ^1^HNMR: (i) MCC, (ii) Tosylated Cellulose (CTOS), (iii) Cell Hyd, (iv) Cell DEA, (v) Cell DETA; (**d**) Zeta potential measurement.

**Figure 2 polymers-12-02634-f002:**
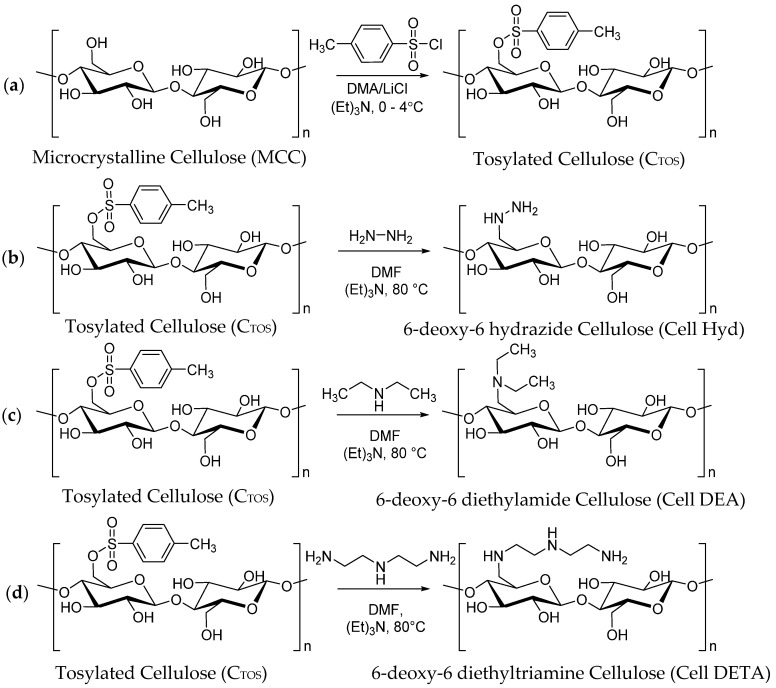
Scheme of Synthesis of Microcrystalline Cellulose (MCC) Derivatives: (**a**) Tosylated cellulose (CTos) synthesis, (**b**) synthesis of 6-deoxy-6-hydrazide Cellulose (Cell Hyd), (**c**) synthesis of 6-deoxy-6-diethylamide Cellulose (Cell DEA), (**d**) synthesis of 6-deoxy-6-diethyltriamine Cellulose (Cell DETA).

**Figure 3 polymers-12-02634-f003:**
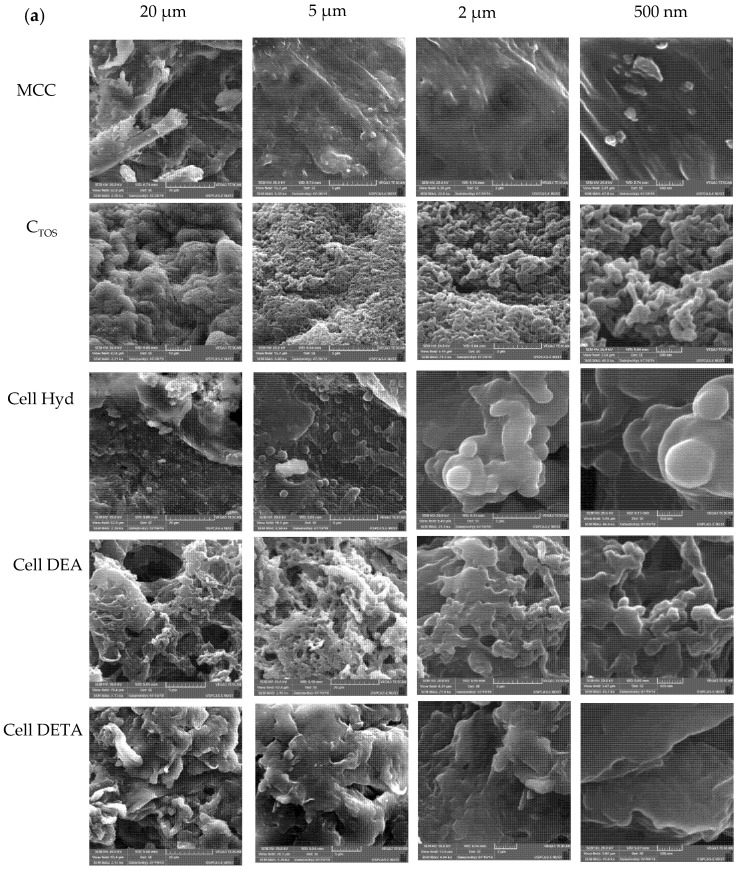
Characterization of Cellulose Derivatives SEM and EDX (Analysis) in solid state of derivatives confirming the synthesis of the derivatives. (**a**) SEM images were taken at 20 µm, 5 µm, 2 µm and 500 µm. (**b**) EXD Analysis (**i**) MCC, (**ii**) C_TOS_, (**iii**) Cell Hyd, (**iv**) Cell DEA, (**v**) Cell DETA.

**Figure 4 polymers-12-02634-f004:**
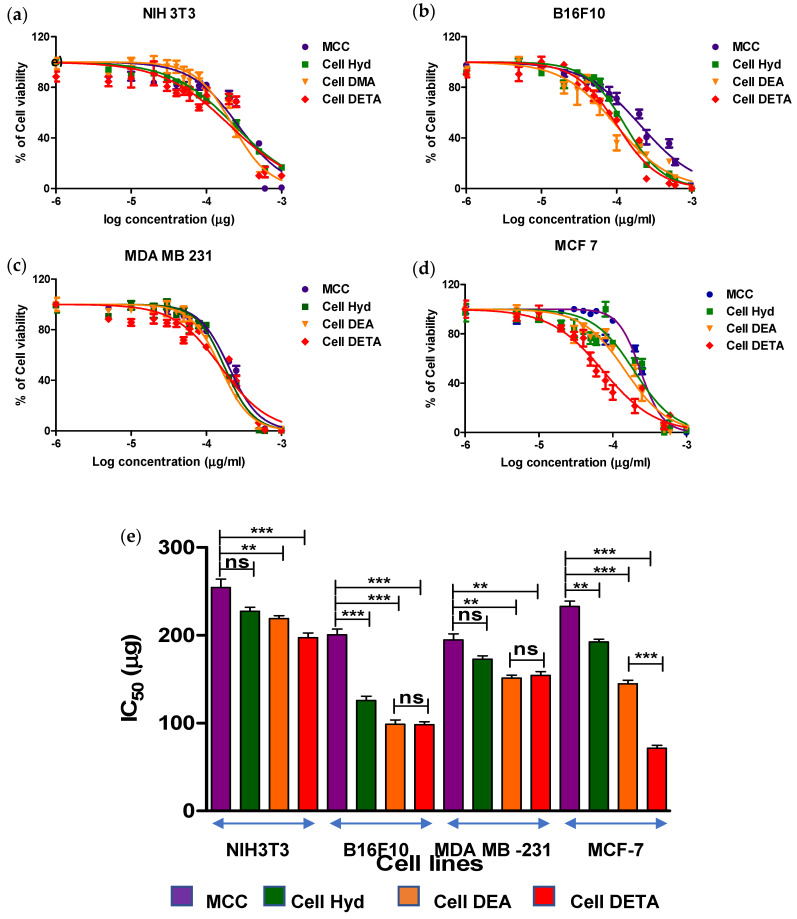
Cell Culture Evaluating the IC_50_ values for the Cellulose and its derivative MCC, Cell Hyd, Cell DEA and Cell DETA ± average standard deviation (* *p* < 0.05, ** *p* < 0.01, and *** *p* < 0.001) against Normal. (**a**) NIH3T3 and Cancer cell lines. (**b**) B16F10. (**c**) MDA-MB-231. (**d**) MCF-7. (**e**) IC50 values graph. (**f**) Swelling ratios of cell derivatives.

**Figure 5 polymers-12-02634-f005:**
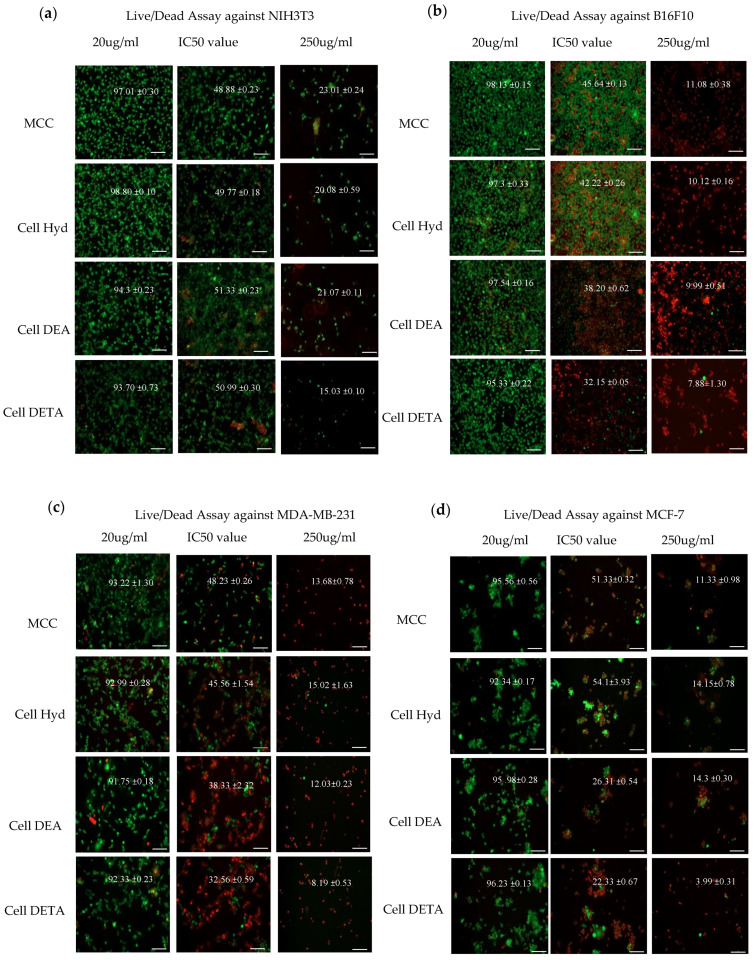
Cell culture live dead assay after treatment with cellulose and its derivatives MCC, Cell Hyd, Cell DEA and Cell DETA, at 3 concentrations after 24 h; against Normal (**a**) NIH3T3 and Cancer cell lines (**b**) B16F10, (**c**) MDA-MB-231, (**d**) MCF-7. Live cells show green fluorescence when stained with calcein-AM, while dead cells show red fluorescence stained with ethidium homodifier. Live cell percentage is shown on the images as total percentage of live cells from four random fields with mean ± standard deviation of 3 replicates of samples. Scale bar is 100 µm.

**Table 1 polymers-12-02634-t001:** Characterization of cellulose amine derivatives elemental analysis) (C, H, N, S).

Cell Lines	Carbon (Cal.)	Carbon (Found)	Hydrogen (Cal.)	Hydrogen (Found)	Nitrogen (Cal.)	Nitrogen (Found)	Sulphur (Cal.)	Sulphur (Found)
MCC	44.31	40.85	6.51	6.34				0.15
C_TOS_	47.59	46.05	5.68	6.05			6.69	4.8
Cell Hyd	42.48	41.34	6.83	6.64	8.26	7.95		0.55
Cell DEA	50.52	46.29	7.95	6.71	3.68	3.25		1.01
Cell DETA	46.82	44.73	7.86	7.24	10.24	9.55		0.77

**Table 2 polymers-12-02634-t002:** Characterization of cellulose amine derivatives zeta potential measurements.

Compounds	Zeta Potential ζ
MCC	−4.24
Cell Hyd	+32.8
Cell DEA	+13.9
Cell DETA	+27

**Table 3 polymers-12-02634-t003:** IC_50_ values for different cell lines.

Cell Lines	NIH3T3	B16F10	MDA-MB-231	MCF-7
MCC	263 ± 13.5	207 ± 12.0	201.6 ± 11.8	238.9 ± 10.9
Cell Hyd	231.9 ±11.5	130.5 ± 11.2	176.5 ± 11.5	195.5 ± 10.8
Cell DEA	222.3 ± 11.3	103.64 ± 9.9	154.6 ± 7.4	148.8 ± 11.2
Cell DETA	202.6 ± 10.35	101.5 ± 6.8	158.5 ± 5.5	74.92 ± 9.9

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
