# Peer review of "Synthesis, Characterization and Cytotoxicity Studies of Aminated Microcrystalline Cellulose Derivatives against Melanoma and Breast Cancer Cell Lines"

_polymers, 2020, doi:10.3390/polym12112634_

Round 1
Reviewer 1 Report
Dear Authors,
the present manuscript entitled "Synthesis, Characterization and Cytotoxicity studies of aminated Microcrystalline Cellulose derivatives against Melanoma and Breast cancer cell lines" is a clear description of derivatization procedures of microcristalline Cellulose.The methodology of the syntheses is described very clearly and in detail and there are many helpful interpretations why which process is performed. What I am still missing is a classification in which respect the represented polymers can be used as delivery platforms. In which way functionalizations can take place here. This also results in a completely new evaluation of the cytotoxicity as it is associated with aminated polymers. Furthermore I miss a discussion about why amination is done by tosylation where there are other possibilities (see specific comments). In the introduction it would be desirable to give a short introduction of alternatives to tosylation like triflate modification or silanization. Basically the manuscript is well conceived and written.
specific comments:
line 53: typo Toslted
line 54: please mention alternate derivatization strategies to make cellulose accesible for SN reactions. Tf or silanisation?
line 70: please Mention clearly to issue of your work
line 90: what is AGU? Please introduce the abreviation
line 232: I have a fundamental question: why do you use tosylation to introduce amino groups intho this polymer. This is also possible by applying I) NaIO4 II)NaBH4 + diethylamine? What is the difference?
line 301: typo "frond"
line 393: typo "applicationit's"
line 504: Maybe I missed it but I would like to have a clear perspective of those aminated dextranes. Ist this the base for further functionalization using aminogroups as good nucleophilic compounds which are prone for further functionalization or should this amino polymer used because of its intrinsic property?
Author Response
Please find response to comments in the attached file.

Reviewer 2 Report
Comments for the Authors
I carefully revised the contribution by Farzana et al. titled “Synthesis, Characterization and Cytotoxicity studies of aminated Microcrystalline Cellulose derivatives against Melanoma and Breast cancer cell lines” which reports the new synthesis of aminated cellulose derivatives for biomedical applications. The Authors found a selective toxic of aminated cellulose for cancer cell lines as compared to normal fibroblast. In particular the mouse skin melanoma (B16F10) are the most sensitive cells to the cytotoxic effects of modified cellulose, followed by human breast adenocarcinoma (MCF-7).
Although the study is well written and organized, I found different substantial flaws thorough out the text in particular in the section reporting the cytotoxic characterization of the materials.
The Authors did not describe the statistical analysis used in this study, the relative paragraph in “M&M” is completely missing. In no one of the figures (4 and 5) reporting the results of the biological characterization of the cellulose are indicated the P values and/or the post-hoc test used to determine the significance. For that reason the Author cannot state, for instance, “….live cell percentage is decreasing significantly….” (Line 482) as well as it is not possible to know if the IC50 (Figure 4) for each cells and materials are significantly different. In term of cytotoxic properties the Authors can not to drive any conclusions unless they repost statistical evidence.
Author Response

(The authors gave the same response as above.)

Reviewer 3 Report
Authors attempted to provide aminated cellulose derivatives for bio-engineering applications and cancer inhibiting studies, but the paper is not acceptable in its present form since it lack of some charactrizations and discussions.
- Native speaking professional editor for language polish is suggested,multiple sentences in the article are inappropriate.E.g: Line 383 “This relatively high cellulose swelling ratio is because of the penetration of the water in intercrystallite amorphous regions of cellulose causing transmutation in the crystal structure of the cellulose from Cellulose I to Cellulose II.”The above sentence is just for enumeration, not all.
- Fig.2 d): Please add the statistical value of CTOS.
- Please provide the sample purity of several derivatives by mass spectrometry, cell death might also casue by the residue of incomplete reaction.
- Fig.5: Please add scale bar,and at the same time, it should be noted at the bottom of the figure what fluorescent markers live and dead cells are, instead of the reader having to correspond with METHODS.
- The current cell experiments are performed with cell lines. If you want to simulate the situation in the body, please select at least one primary cell for verification.
Author Response

(The authors gave the same response as above.)

Reviewer 4 Report
The authors report synthesizing aminated cellulose derivates that are selectively toxic to the cancer cells but not harmful to the normal cells (fibroblasts) up to a certain concentration.
There are a few queries/comments to be addressed:
- Lines 196-231: What is the significance of the degree of substitution and why the value of 0.48 is selected? The paper states that the tosylated cellulose prepared is DMF soluble, which seems to be dependent on the DS value. Why is this important?
- Lines 233-235: Does the amount of amine not determine the aminated %? It is just mentioned that excess amount of amine is used to prevent side reaction and crosslinking but how much in excess is really excessive?
- Much of the FTIR data are repeated both in section 2.2.2. and 3.2. Section 3.2 could be shortened.
- Lines 394-395: It is good to state those "very less study", their results and how this study is novel (new results) compared to the previous studies?
- Is the data presented in Figure 4(e) statistically significant? Where are the error bars and p values? If no statistical data is obtained, then the claim that the material developed is toxic to the cancerous cells but not normal cells wouldn't be valid. This is a major point to address.
- What is the hypothesis behind the observation that material developed is toxic to the cancerous cells but not normal cells wouldn't be valid?
Author Response

(The authors gave the same response as above.)

Round 2
Reviewer 2 Report
The Authore have coped the requests. I think the manuscript is substantially improved.
Reviewer 3 Report
The manuscript has been carefully revised and is almost ready for publication.
Reviewer 4 Report
All the comments are suitably addressed by the authors. It is recommended that the authors incorporate the responses to comments 4 and 6 in the manuscript.